# Microparticles in Human Perspiration as an Inflammatory Response Index

**DOI:** 10.3390/diagnostics14121293

**Published:** 2024-06-19

**Authors:** Zuha Imtiyaz, Veena M. Bhopale, Awadhesh K. Arya, Abid R. Bhat, Stephen R. Thom

**Affiliations:** Department of Emergency Medicine, University of Maryland School of Medicine, Baltimore, MD 21201, USA; zimtiyaz@som.umaryland.edu (Z.I.); vbopale@som.umaryland.edu (V.M.B.); aarya@som.umaryland.edu (A.K.A.); arbhat@som.umaryland.edu (A.R.B.)

**Keywords:** decompression sickness, exercise, exosomes, interleukin (IL)-1β

## Abstract

A blood component analysis is an early step for evaluating inflammatory disorders, but it can be unfeasible in some settings. This pilot study assessed whether extracellular vesicle (EV) changes in perspiration are parallel to those occurring in blood as an alternative or complementary option to diagnose an inflammatory response. In parallel studies, EVs were analyzed in perspiration and blood obtained before and after five self-contained underwater breathing apparatus (SCUBA) divers at the National Aquarium in Baltimore performed a dive to 3.98 m of sea water for 40 min, and five non-divers performed an exercise routine at ambient atmospheric pressure. The results demonstrated that microparticles (MPs) are present in perspiration, their numbers increase in the blood in response to SCUBA diving, and the interleukin (IL)-1β content increases. In contrast, while blood-borne MPs became elevated in response to terrestrial exercise, no statistically significant increases occurred in perspiration, and there were no changes in IL-1β. There were no statistically significant elevations in the exosomes in perspiration or blood in response to SCUBA diving and few changes following terrestrial exercise. These findings suggest that an MP perspiration analysis could be a non-invasive method for detecting inflammatory responses that can occur due to the oxidative stress associated with SCUBA diving.

## 1. Introduction

Analyzing blood components is a central aspect to diagnosing inflammatory disorders. However, in emergency settings or at remote out-of-hospital locations, phlebotomy may not be feasible. We hypothesized that skin perspiration could be used as an alternative for evaluating inflammatory disorders where extracellular vesicles (EVs) have been shown to be elevated in blood. Perspiration is a body fluid obtainable in a non-invasive fashion. Antibodies, cytokines, nucleic acids, antimicrobial/defense proteins, and protein metabolites are present in human perspiration [1,2,3]. There is precedence for using some of these substances to assess disease states, although there is no direct communication between perspiration and blood components [2,4,5,6]. Perspiration also contains endosomal-derived exosomes, ~20–100 nm diameter EVs, and some particle characteristics that represent parallel findings to those for blood in response to high-intensity exercise [7].

We and others have shown that elevations in microparticles (MPs), one type of EV, occur in blood in response to varied conditions such as acute carbon monoxide poisoning, decompression sickness (DCS), diabetes mellitus, opioid use disorder, and traumatic brain injury [8,9,10,11,12,13]. MPs are 0.1–1 µm diameter vesicles generated by an outward budding of the plasma membrane, and some carry pro-inflammatory proteins [8,9,10,11,12,13,14]. Leukocytes are responsible for producing many of these MPs. We reasoned that, because skin keratinocytes and Langerhans cells carry out many of the same immune-related functions as blood-borne leukocytes, MPs may be generated in perspiration [15,16]. 

This was a pilot study designed to assess several questions. First, as MPs have not been reported in human perspiration, the first goal was to assess their presence. We also wanted to determine whether MPs in perspiration would correlate with the blood-borne MP changes that we have reported in prior publications, that is, MPs expressing surface proteins that are specific to formed blood elements such as neutrophils and platelets, as well as those expressing proteins specific to the endothelium and microglia [8,9,10,11,12,13]. To accomplish this, we established a protocol with research subjects undertaking underwater diving with a self-contained underwater breathing apparatus (SCUBA). Elevations in inflammatory MPs occur in response to this activity [14,17]. That is, blood-borne changes occur without triggering DCS. Therefore, studies can be ethically undertaken to compare the EVs in the perspiration and blood of human SCUBA divers. An additional, practical aspect to this study is that there are times when diagnosing DCS can be difficult [18]. Therefore, should MP changes be found in perspiration, it may have future clinical value. As a comparison to the responses of divers, studies were also performed to assess whether similar changes occurred in response to exercise at ambient air pressure.

## 2. Materials and Methods

### 2.1. Study Subjects

All the subjects gave their informed consent for inclusion before they participated in this study. The enrolled subjects were five SCUBA divers aged 42.4 ± 16.3 years old (mean ± SD), including three women and two men, who were employed by the National Aquarium in Baltimore, MD. Five additional subjects aged 32.8 ± 5.4 years old, including two women and three men, were enrolled for providing blood and perspiration samples prior to and following terrestrial exercise.

### 2.2. Protocol

All the procedures were completed in accordance with the Declaration of Helsinki and approved by the Institutional Review Board of the University of Maryland (HP-00095857). The perspiration samples to be used as a control were obtained from research subjects wearing collection devices on their skin while conducting their normal daily activities for ~5 h. Perspiration was then collected after the research subjects performed a SCUBA dive to 3.98 m of sea water (msw) for 40 min or after terrestrial exercise, which was a combination of a cardio workout (2–3-mile indoor run) and strength training with weights for at least 1 h. Blood samples were obtained from the research subjects immediately before and after the SCUBA dive or terrestrial exercise.

### 2.3. Materials

The chemicals were purchased from Sigma-Aldrich (St. Louis, MO, USA) unless otherwise noted. Laurdan (1-[6-(dimethylamino)-2-naphthalenyl]-1-dodecanone, cat # 7275) was purchased from Tocris Biotech (Minneapolis, MN, USA). The antibodies were validated by commercial sources with the indicated host specificity, and the data sheets of each product were supplied as follows. Annexin-binding buffer and the following antibodies were purchased from BD Pharmingen (San Jose, CA, USA): PE-conjugated annexin V (cat # 556421), PE-conjugated anti-human CD18 (cat # 555924), APC-conjugated anti-human CD41a (cat # 559777), and FITC-conjugated anti-human CD15 (cat # 555401). BV421-conjugated anti-human CD66b (cat # 347201) and BV510-conjugated anti-human CD146 were purchased from BD Optibuild (San Jose, CA, USA) and BD Horizon (San Jose, CA, USA), respectively. PerCP-Cy5.5-conjugated anti-human CD41a (cat # 340931), percp5.5-conjugated anti-human CD14 (cat # 550787), and FITC-conjugated anti-human myeloperoxidase (MPO, cat # 340655) were purchased from BD Biosciences (San Jose, CA, USA). AlexaFluor488-conjugated anti-human TMEM119 (cat # ab225494) was purchased from Abcam (Boston, MA, USA). FITC-conjugated anti-human thrombospondin 1 (sc-393504) and AF647-conjugated anti-human NKCC1 (cat # sc-514858) were purchased from Santa Cruz Biotechnology (Dallas, TX, USA). PE-Vio770-conjugated anti-human CD-18 integrin (cat # 130-11-217) was purchased from Miltenyi Biotec (Bergisch Gladbach, Germany). The antibodies purchased from Biolegend (San Diego, CA, USA) included the following: AlexaFluor647-conjugated anti-human CD63 (cat # 353016), PercpCy5.5-conjugated anti-human CD81 (cat # 349520), BV421-conjugated anti-human CD66b (cat # 347201), and FITC-conjugated anti-human CD1a (cat # 300128).

### 2.4. Blood Collection

Blood was collected in Cyto-Chex^®^ BCT (Streck, La Vista, NE, USA) test tubes following a sterile, standard procedure [17].

### 2.5. Perspiration Collection

Perspiration collection involved two procedures. The research subjects wore a perspiration collection system known as a Discovery patch^®^ (Epicore biosystems, Cambridge, MA, USA) designed specifically for perspiration collection. It is a wearable microfluidic patch that collects eccrine perspiration directly from the skin. However, because we found that the collections were variable, we instituted a second procedure performed in parallel with the patch. We found that reliable collections were achievable by placing a 1 cm^2^ cotton felt patch that was adhered to the skin using Tegaderm™ film (3M Healthcare, St. Paul, MN, USA). Because the felt patches sometimes absorbed unwanted debris, the liquid perspiration was passed through a 20–40 µm filter prior to the analysis. The preliminary studies established that there were no significant differences in the EV counts using perspiration collected with the Discovery patch^®^ or cotton felt, so the samples were combined for standard analyses. We also found that some of the SCUBA divers generated scant perspiration while immersed in 23.9 °C aquarium water wearing standard wet suits. This was remedied when the divers switched to dry suits that prevented water contact with their skin.

The research subjects were advised to not use any moisturizers on the day of perspiration collection, as moisturizers could interfere with the patch adhesion. The lower back of all the volunteers was selected as the site for the Discovery patch^®^ and 1 cm^2^ cotton felt patch placement. The lower back area was thoroughly cleaned with alcohol swabs. Discovery patches^®^ were placed on the surface whilst maintaining palm pressure to ensure secure adhesion. The cotton felt patches were placed adjacent to the Discovery patches^®^. For the diving volunteers, both of the perspiration collection devices were covered with Tegaderm™ film (3M Healthcare, St. Paul, MN, USA) to avoid potential interference should aquarium water leak into their diving suits. After the completion of the designated activities, the perspiration was transferred from the patches to sterile vials. For perspiration extraction from the Discovery patches^®^, a proprietary extraction apparatus was utilized. The perspiration was extracted from the felt patch by placing it in a 3 mL syringe and depressing the plunger to force out the liquid.

### 2.6. Extracellular Vesicle Isolation

The EVs were isolated from blood by following published procedures and were characterized according to prior publications based on their size, Western blots, and electron microscopy [8,9,10,11,12,13,14,17,19]. In brief, the blood was centrifuged for 5 min at 1500× *g*, followed by the addition of EDTA to the supernatant to achieve a final concentration of 12.5 mmol/L and then centrifugation at 15,000× *g* for 30 min. The supernatant was centrifuged at 21,000× *g* for 30 min to separate the MPs (in a pellet) from the exosomes, which remained in the supernatant. The perspiration samples were processed similarly after adding a commercial fixative (100 µL/mL Caltag Reagent A fixation medium, Invitrogen, Carlsbad, CA, USA).

### 2.7. MP Analysis

The MP analysis was performed using an 8-color, triple-laser MACSQuant (version 2.13.3, Miltenyi Biotec Corp., Auburn, CA, USA) flow cytometer. The MPs and subtypes were stained with antibodies to identify the positive vs. negative particles, as described in a previous study [17]. 

### 2.8. Exosome Analysis

The exosome analysis was performed by followed published procedures [17]. In brief, 5 µL of the supernatant from a plasma or perspiration sample after 21,000× *g* centrifugation was diluted in 100 µL of PBS. The samples were then incubated with dyes (2.5 µmol PKH67 and/or 2.5 µmol Laurdan) and antibodies to CD63 and CD81 for 30 min, followed by an analysis using an ImageStream^®^X Mk II: Imaging Flow cytometer. FluoSphere™ carboxylate-modified microspheres (Thermofisher; 20 and 100 nm in diameter) were used to provide size bracketing. After applying the compensation matrix with the bright field off and all the channels enabled, single-color compensations were set for each color, with gates set to detect particles between 20 and 100 nm in diameter. 

### 2.9. IL-1beta

The blood EVs were analyzed by following published procedures using a commercial human-specific ELISA kit (eBioscience, San Diego, CA, USA) that detected pro- and mature forms of IL-1β. The measurements were made using the plasma supernatant (MPs) and pellets (exosomes) after sample centrifugation at 21,000× *g*, as described above for EV isolation. Because IL-1β was not detected by ELISA in the perspiration EVs, Western blot analyses were performed so that higher amounts of protein could be used in the analyses. Whole perspiration samples were centrifuged at 100,000× *g* for 60 min, and then the pellet was suspended in RIPA buffer and subjected to Western blot analyses by following published procedures [14].

### 2.10. Statistical Analysis

The results are expressed as the mean ± standard deviation (SD), and the analyses were performed using SigmaStat (v15, Jandel Scientific, San Jose, CA, USA). Data normality was assessed using the Shapiro–Wilk test. The details of the statistical analysis for each assay are provided in the figure legends. Single-group (pre- and post-activity) comparisons were evaluated by a paired *t*-test. For multiple group comparisons, a one-way analysis of variance (ANOVA) was used. For all the studies, the results were deemed statistically significant if *p* < 0.05.

## 3. Results

### 3.1. Correlation between Perspiration and Blood MPs

The results for the MPs in blood and perspiration from SCUBA divers are shown in Figure 1A,B. Statistically significant increases post-SCUBA diving were found for the total MP number and for several MP sub-groups expressing specific proteins, which is consistent with previous studies [14,17]. The within-subject variation for duplicate samplings was less than 2%, whereas the variation among the five divers is shown in the figure. Significant increases were found for blood-borne MPs expressing neutrophil CD66b proteins, endothelial cell CD146, and platelet CD41a, but not for microglial TMEM119. MPs expressing thrombospondin-1 (TSP) and filamentous (F-) actin, proteins previously found to directly contribute to inflammatory responses post-SCUBA diving, were also significantly increased. As with prior studies, the total number of MPs tallied among all the subgroups exceeded 100%, thus indicating that MPs share surface proteins [14,17,20]. In the perspiration samples, statistically significant increases post-SCUBA diving were found for MPs expressing the epithelial cell β5 integrin, Langerhans cell CD-1a, apocrine gland CD-15, the pan-leukocyte protein CD-14, eccrine gland NKK1, and F-actin (Figure 1B).

A second set of studies was conducted with the individuals performing terrestrial exercise. As shown in Figure 2A, increases in the total MPs and most MP sub-groups were found in the post-exercise blood. There were no significant differences pre- versus post-exercise in the total number of perspiration MPs. The fractions of some MP sub-groups expressing proteins specific to several cells were significantly lower post-exercise.

### 3.2. Correlation between Perspiration and Blood Exosomes

The exosomes in the perspiration and blood from SCUBA divers were enumerated, and the results are shown in Figure 3A,B. There were no significant differences in the blood exosomes pre- versus post-SCUBA diving, and the post-SCUBA diving perspiration exhibited decreases in several exosome sub-groups.

The results for the subjects who performed terrestrial exercise are shown in Figure 4A,B. Again, there were virtually no statistically significant differences in the blood exosomes following terrestrial exercise. With the high inter-subject variability, there were no significant differences in pre-activity blood exosomes among the five subjects who performed SCUBA diving and the five who performed terrestrial exercise. The pattern of exosomes in perspiration pre- versus post-terrestrial exercise followed no uniform pattern of changes, with some groups increasing while others decreased. 

### 3.3. IL-1B in EVs

IL-1β is synthesized without a leader peptide, so that cell export requires its packaging into vesicles such as exosomes or MPs [21,22]. As with prior studies, the content of IL-1β in post-diving blood MPs was significantly greater than that in pre-diving MPs [14,17]. The IL-1β content pre-SCUBA dive was 8.7 ± 3.0 pg/million MPs and that post-dive was 30.5 ± 13.6 pg/million MPs (*p* = 0.008). However, the IL-1β content in blood MPs did not differ pre- versus post-terrestrial exercise. The MPs from the subjects pre-exercise had an IL-1β content of 9.9 ± 4.4 pg/million MPs, and post-exercise, the content was 10.9 ± 6.6 pg/million MPs (NS). Again, consistent with a prior study, there was no detectable IL-1β in blood exosomes among the samples from the subjects who performed SCUBA dives or terrestrial exercise [14,17].

We were unable to detect IL-1β in perspiration EVs by ELISA. We interpreted this outcome as suggesting that the cytokine concentration was too low with the relatively high background of other proteins. As an alternative approach, we subjected the perspiration samples to ultracentrifugation and probed for IL-1β by Western blotting. Again, no IL-1β was detected in the exosomes, in the MPs in perspiration pre- SCUBA diving, or in the samples pre- and post-terrestrial exercise. However, IL-1β was detected in two post-SCUBA diving perspiration MP samples. Based on standards run on the same Western blots, the IL-1β concentrations were estimated to be 0.16 and 0.17 pg/million MPs, which was approximately 0.5% of the concentration of post-SCUBA diving blood MPs.

## 4. Discussion

The results demonstrate a close relationship between MP elevations in the blood and perspiration in response to SCUBA diving. As it can be difficult at times to diagnose DCS, this finding suggests that there may be merit to developing an analytical platform designed to provide a non-invasive assessment of inflammatory MPs that could be used in remote out-of-hospital environments. Interestingly, while there were elevations in the MPs in blood related to terrestrial exercise, they were not mirrored by changes in the perspiration. In this regard, it is important to note that we have previously reported that the exertion associated with swimming on the water surface does not trigger elevations in blood MPs [23]. It is unknown whether physiological differences inherent with the type of exercise or other factors such as thermal stress are responsible for the different MP responses between swimming and terrestrial exercise.

We hypothesize that the mechanism for MP elevation in perspiration following SCUBA diving is the same as that for MP production by leukocytes in the blood. Validation will require additional study. The elevated partial pressures of gases such as nitrogen that occur while SCUBA diving trigger oxidative stress in leukocytes that prompt MP production, the formation of the NLRP3 inflammasome, and increased IL-1β concentrations in MPs [24]. Skin keratinocytes and Langerhans cells carry out many of the same immune-related functions as blood leukocytes [15,16]. Therefore, these skin cells may respond to elevated gas partial pressures in the same manner. If so, this mechanism may be the basis for so-called “skin bends”, characterized by a peculiar type of edema, a rash, pruritis, and non-dermatomal paresthesia that occur in as many as 50% of DCS cases [18]. The SCUBA dive performed by the research subjects in this investigation was a nominal stress, so further research is needed involving more provocative diving to investigate this question. 

As was discussed in the Introduction, elevations in blood-borne MPs are seen in association with numerous pathological events. The biochemical mechanisms triggered by carbon monoxide, hyperglycemia, and hypercarbia that lead to MP formation are likely shared by many cells [10,25,26]. Hence, it is feasible that monitoring MP changes in perspiration in response to conditions other than SCUBA diving could be useful for assessing immune responses that contribute to pathological changes. We think it is notable that perspiration MPs with the manifestation of surface proteins characteristic of a wide variety of cell types were all increased due to SCUBA diving. Hence, cells with differing phenotypes may respond to the stress of diving in a similar fashion. 

Finally, it is notable that we did not observe predictable patterns of exosome changes in the blood or perspiration following SCUBA diving or terrestrial exercise. Exosomes are generated by an entirely different mechanism than MPs, so discrepant responses between these EVs is not surprising [27]. Interestingly, we observed markedly different responses among the exosome sub-types; for example, some increased while others decreased in response to terrestrial exercise. This suggests that the processing and release of exosome sub-types differ among cells. The technology for evaluating exosome sub-types is still evolving and questions related to their physiology will require substantial additional research.

## Figures and Tables

**Figure 1 diagnostics-14-01293-f001:**
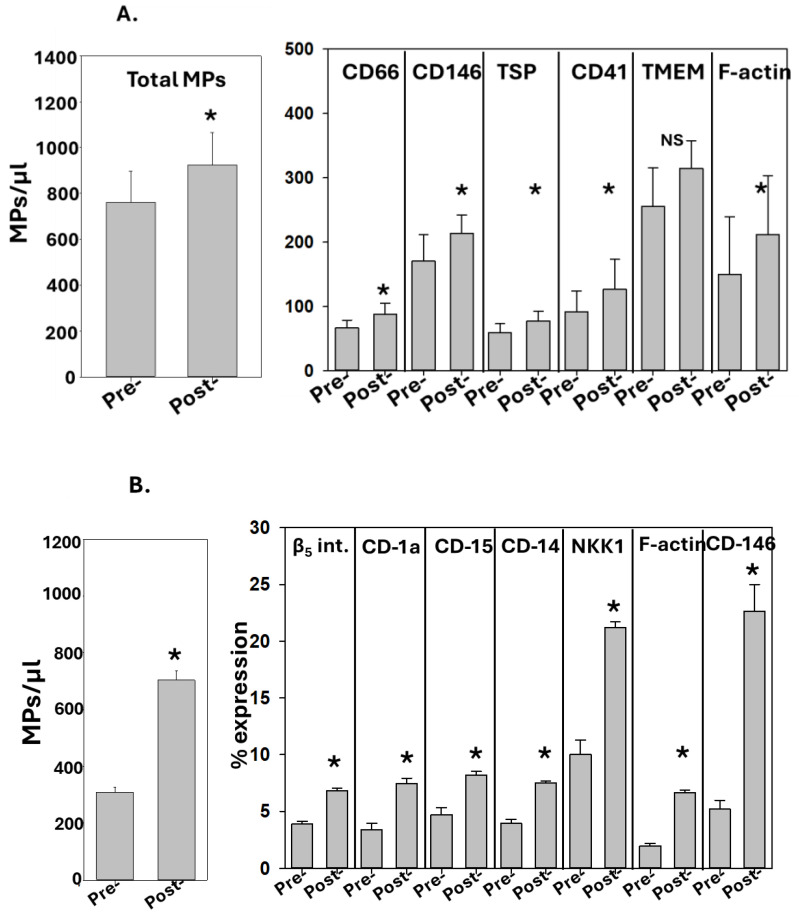
**MPs in diving subjects**. The data show the MPs in (**A**) blood and (**B**) perspiration. Blood and perspiration were collected before and after a dive to 3.98 msw for 40 min, as described in the Methods. The sub-groups for blood MPs are represented as the number of MPs/µL and they display a significant increase post-dive as compared to pre-dive (mean ± SD, *n* = 5, * *p* < 0.05, paired *t*-test). The sub-groups for perspiration MPs are shown as the % of MPs expressing proteins specific to alternative cell types, as described in the text, with significant increases post-dive (mean ± SD, *n* = 5, * *p* < 0.001, paired *t*-test; NS indicates not statistically significant).

**Figure 2 diagnostics-14-01293-f002:**
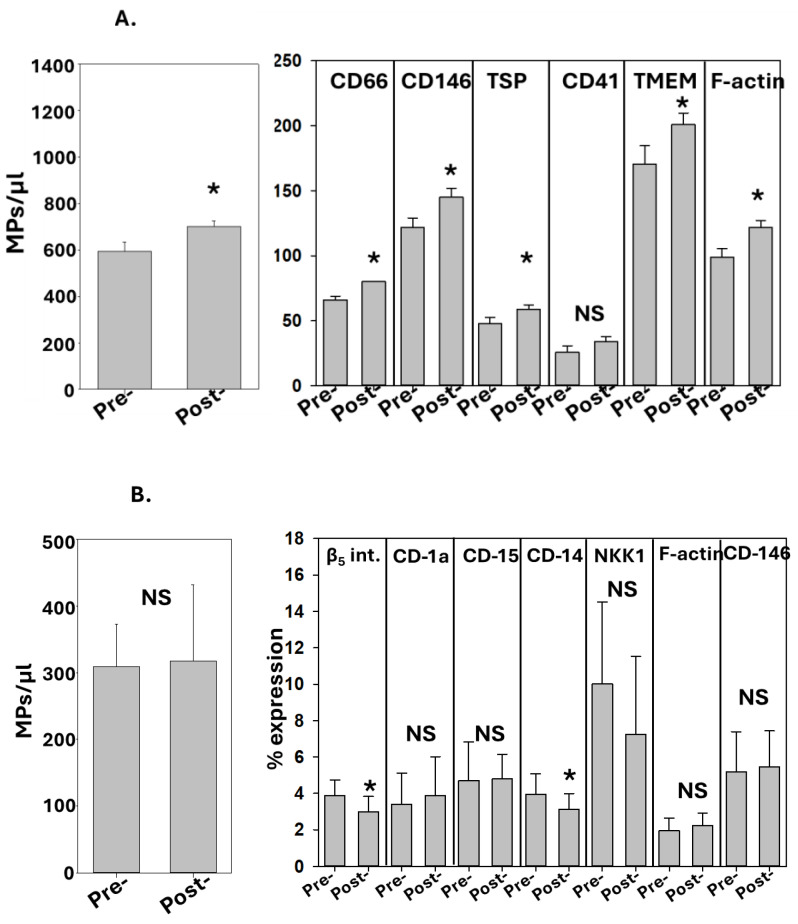
**MPs in non-diving subjects.** The data show the MPs in (**A**) blood and (**B**) perspiration. Blood and perspiration were collected before and after a 60 min terrestrial workout, as described in the Methods. The sub-groups for blood MPs are represented as the number of MPs/µL, where significant increases were seen for all the sub-groups post-exercise except for CD41 (mean ± SD, *n* = 5, * *p* < 0.05, paired *t*-test). The sub-groups for perspiration MPs are represented as the % of MPs expressing proteins specific to alternative cell types, as described in the text. Significant changes post-exercise are as shown (mean ± SD, *n* = 5, * *p* < 0.05, paired *t*-test; NS indicates not statistically significant).

**Figure 3 diagnostics-14-01293-f003:**
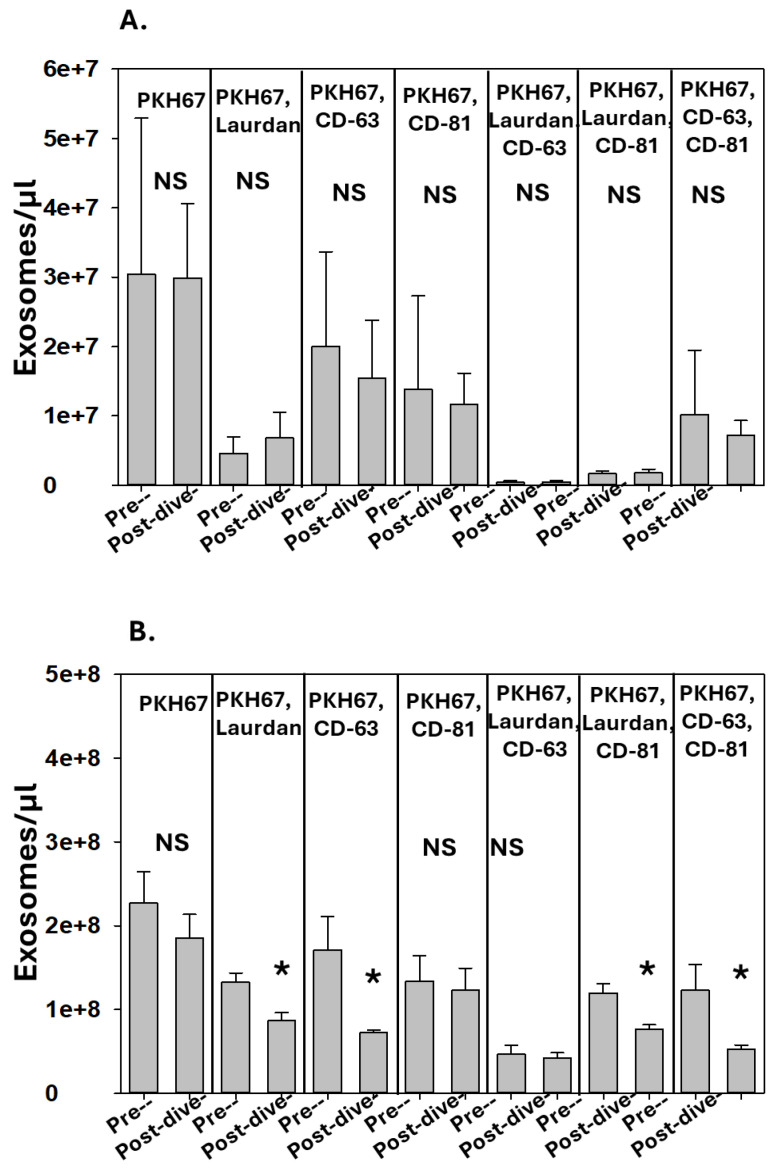
**Effect of diving on exosomes**. Data show number of exosomes/µL in (**A**) blood and (**B**) perspiration. No statistically significant changes in blood occurred post-diving. Significant changes in perspiration post-diving are as shown (mean ± SD, *n* = 5, * *p* < 0.05, paired *t*-test; NS indicates not statistically significant).

**Figure 4 diagnostics-14-01293-f004:**
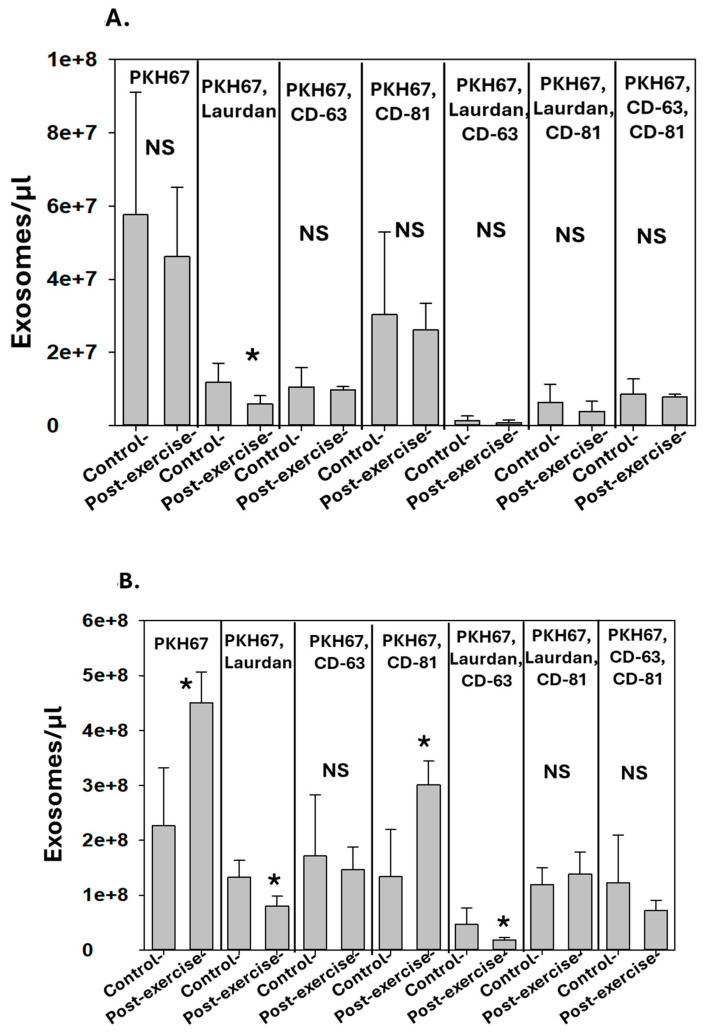
**Effect of terrestrial exercise on exosomes**. Data show number of exosomes/µL in (**A**) blood and (**B**) perspiration. Significant changes are as shown (mean ± SD, *n* = 5, * *p* < 0.05, paired *t*-test; NS indicates not statistically significant).

## Data Availability

All data and materials from this project are available upon request.

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
