# Peer review of "Microparticles in Human Perspiration as an Inflammatory Response Index"

_diagnostics, 2024, doi:10.3390/diagnostics14121293_

Round 1

Reviewer 1 Report

Comments and Suggestions for Authors

In this case study, Authors showed Microparticles in human perspiration as an inflammatory response index. These findings suggest that MPs analysis of perspiration can be a non-invasive method to detect inflammatory responses as occur due to the oxidative stress associated with SCUBA diving. This case report aims to contribute to the current understanding of This is a well-written manuscript, the author made a great effort, but I suggest some change that must be made to the text, to contribute to a better understanding of the points they are trying to make.

comments

1.     In figure 1A, 2A and 2B, if you compare standard deviation bars. These bars are more in 1A making it NS. Please explain.

2.     In line 196, 197 and 198. In figure 2B also bars making MPs NS.

3.     Even in all figures, when author is saying NS bars are so high, So much deviation in replicates. Please explain.

4.     In line 209 and 210. As you said, “With the high inter-subject variability, there were no significant differences pre-activity blood exosomes between the 5 subjects”. Please take more subject, so that difference between subject will be less, please explain.

Author Response

  1. In figure 1A, 2A and 2B, if you compare standard deviation bars. These bars are more in 1A making it NS. Please explain. Please see below.
  2. In line 196, 197 and 198. In figure 2B also bars making MPs NS. Please see below.
  3. Even in all figures, when author is saying NS bars are so high, So much deviation in replicates. Please explain. Please see below.
  4. In line 209 and 210. As you said, “With the high inter-subject variability, there were no significant differences pre-activity blood exosomes between the 5 subjects”. Please take more subject, so that difference between subject will be less, please explain. Please see below.

Response to all of reviewer #1 comments: The reviewer has pointed out the variability of the data in our pilot study. We can offer several ideas that may explain some findings. Regarding variability within the various figures, some differences are likely to be based on the fluid being sampled with observations between blood (Figure 1 A) and sweat (Figure 1 B) among the 5 SCUBA divers and the non-diving ‘terrestrial exercise’ subject’s blood (Figure 2A) and sweat (Figure 2 B). In Figure 1A the only sub-group that was not statistically significant was TMEM-expressing particles, which we interpret as indicating that the nominal pressure exposure was insufficient to trigger a CNS-mediated response, contrary to what we have seen in more provocative pressure exposures. In comment #3 the reviewer appears to be questioning the degree of variability within the replicate sampling. To clarify our findings, we have added a sentence to results as follows: “The within-subject variation for duplicate samplings was less than 2% whereas the variation among the 5 divers is as shown.” The magnitude of particle elevations in sweat (Figure 1B) was greater than we saw in the blood, with somewhat less variability. This finding underscores the possibility of using perspiration to assess inflammatory response in a non-invasive manner. While terrestrial exercise did stimulate an elevation of MPs in blood, no change was observed in perspiration. Moreover, the post-exercise perspiration EVs values were often lower than pre-exercise.

As suggested by the reviewer, one approach to further assessment of these findings is to add more subjects. Respectfully, we disagree that simply adding more will improve the nature of our pilot project. The erratic pattern of MPs changes in perspiration post-terrestrial exercise, and exosomes in response to both SCUBA diving and terrestrial exercise (some values increasing, some decreasing) suggests that there is no uniform response. We believe we have established the feasibility of using sweat MPs measurements as an index of inflammation. Despite the small number of subjects, consistent and statistically significant elevations of sweat and blood MPs were observed. Rather than merely adding more subjects, we believe a greater value going forward is to focus on more provocative diving where the risk of an adverse effect (in this case, decompression sickness) is higher.

We thank the reviewers for their insightful comments and the changes that were prompted have improved our paper.

Reviewer 2 Report

Comments and Suggestions for Authors

Studying the possibility of using extracellular vesicles as diagnostic markers for a variety of diseases is one of the most promising and studied areas of modern science. Extracellular vesicles are found in almost all biological fluids of the body. However, sweat has so far been outside the scope of these studies. The work compares the population of extracellular vesicles isolated from sweat and vesicles isolated from blood.

There are the following comments and questions regarding the work.

The introduction lacks information about the relevance of the work: is diagnosing inflammatory diseases a problem? Is diagnosing DCS a problem? 

In the introduction, it is necessary to justify the choice of people performing physical exercises as a comparison group when searching for markers of inflammatory diseases.

The term "blood-borne" is questionable. I am not a native English speaker, however, when talking about vesicles derived from or circulating in the blood, it is more appropriate to use the term "blood-derived" or simply "blood".

To denote the liquid being studied, the word sweat or perspiration is used. FOR the convenience of the reader, it is better to use one term. Which of them most adequately reflects the liquid that was studied?

Was the method for isolating microparticles and exosomes developed by the authors or taken from the literature? On line 135 it is necessary to provide a link to the work where the isolation method is verified according to the requirements of the Society of Extracellular Vesicles, that is, using flow cytometry, Western blotting and electron microscopy. Such a simple and effective method for separating exosomes and microparticles causes both doubt and admiration in the reviewer.

A description of the process for rinsing sweat samples from cotton patches should be added to the Materials and Methods. Was there any loss of part of the sample at this stage?

The authors suggest that the source of microparticles in sweat are Langerhans cells and skin keratinocytes. Perhaps, to test this theory, it makes sense to check washes from the surface of the skin for the presence of microparticles?

Author Response

The introduction lacks information about the relevance of the work: is diagnosing inflammatory diseases a problem? Is diagnosing DCS a problem? Response: Difficulty with making the diagnosis of DCS was mentioned in the Discussion of our initial submission. In response to the reviewer’s comment, we have added information to the Introduction section to highlight the issue that diagnosing DCS can be difficult. Often it is based solely on clinical history, and laboratory-based objective testing in remote locations is at times impossible.

In the introduction, it is necessary to justify the choice of people performing physical exercises as a comparison group when searching for markers of inflammatory diseases. Response: We have added to the Introduction that our motivation for including a ‘terrestrial exercise’ group to the study was to offer a comparison to the stresses of SCUBA diving. The individuals were chosen randomly for both groups.

The term "blood-borne" is questionable. I am not a native English speaker, however, when talking about vesicles derived from or circulating in the blood, it is more appropriate to use the term "blood-derived" or simply "blood". We understand the reviewer’s concern, but believe our terminology is most accurate. The reason we investigated sub-sets of MPs expressing proteins specific to alternative cells is to obtain information on their sources. To use alternative terms such as ‘blood-derived’ or just ‘blood’ suggests that the MPs are solely generated by blood cells, which is not the case.

To denote the liquid being studied, the word sweat or perspiration is used. FOR the convenience of the reader, it is better to use one term. Which of them most adequately reflects the liquid that was studied? In deference to the reviewer’s comment, perspiration has now been used throughout the manuscript.

Was the method for isolating microparticles and exosomes developed by the authors or taken from the literature? On line 135 it is necessary to provide a link to the work where the isolation method is verified according to the requirements of the Society of Extracellular Vesicles, that is, using flow cytometry, Western blotting and electron microscopy. Such a simple and effective method for separating exosomes and microparticles causes both doubt and admiration in the reviewer.

We have added citations to our prior papers where methodology for microparticle and exosome isolation and analyses were described in detail. The International Society for Extracellular Vesicles (ISEV) has acknowledged that there is no firmly established approach for EVs analysis and stated that no current isolation protocol can purify based on biogenic origin or size alone. In our various publications now cited in this manuscript we have described results based on flow cytometry, Western blotting, and electron microscopy. The ISEV recommends that EVs are described based on physical characteristics, biochemical characteristics and cell of origin or stimulus condition. As with all of our prior publications in the current study MPs were characterized based on size, annexin V staining (presence of phosphatidylserine in the membrane) and well as an array of surface proteins. Exosomes were characterized based on size, some tetraspanin content and we also further evaluated lipid character based on several other agents (as we have described in detail in citation #17). 

A description of the process for rinsing sweat samples from cotton patches should be added to the Materials and Methods. Was there any loss of part of the sample at this stage? Information was added to the Methods section.

The authors suggest that the source of microparticles in sweat are Langerhans cells and skin keratinocytes. Perhaps, to test this theory, it makes sense to check washes from the surface of the skin for the presence of microparticles?

The presence of proteins on MPs that are relatively specific to Langerhans cells and keratinocytes prompted our suggestion that MPs originated from these cells. Respectfully, we do not believe washing the skin surface will improve confidence that MPs arose from these cells. More highly sophisticated studies with isolated cells under alternative stresses ex vivo would be insightful and add mechanistic information, but these experiments are beyond the scope of our pilot study.

We thank the reviewers for their insightful comments and the changes that were prompted have improved our paper.

Round 2

Reviewer 1 Report

Comments and Suggestions for Authors

Thank you for your revisions.

Author Response

Thank you. Reviewer #1 had no additional issues.

Reviewer 2 Report

Comments and Suggestions for Authors

Dear authors, thank you for the changes made. Thanks to them, the text, in my opinion, has become much more readable, the problem at which the research is aimed at solving is more clearly expressed, and the essence and prospects are clear. I only have one request left. I completely agree with your explanation of the use of the term blood-borne and I only ask that you provide in the text a definition of exactly which microparticles you understand by blood-borne.

Author Response

The reviewer has asked that we provide in the text a definition of exactly which microparticles we understand by blood-borne.

We have added information in the third paragraph of the Introduction. The second and third sentences now read: ”We also wanted to determine whether MPs in perspiration would correlate with blood-borne MPs changes we have reported in prior publications. That is, MPs expressing surface proteins specific to formed blood elements such as neutrophils and platelets, as well as those expressing proteins specific to endothelium and microglia [8-13].”